# Regulatory Effect of Methylation of the Porcine *AQP3* Gene Promoter Region on Its Expression Level and Porcine Epidemic Diarrhea Virus Resistance

**DOI:** 10.3390/genes11101167

**Published:** 2020-10-06

**Authors:** Jia-Yun Wu, Fang Wang, Zheng-Chang Wu, Sheng-Long Wu, Wen-Bin Bao

**Affiliations:** 1Key Laboratory for Animal Genetics, Breeding, Reproduction and Molecular Design of Jiangsu Province, College of Animal Science and Technology, Yangzhou University, Yangzhou 225009, China; YZUwujiayun@163.com (J.-Y.W.); wangfangRD@163.com (F.W.); zcwu@yzu.edu.cn (Z.-C.W.); slwu@yzu.edu.cn (S.-L.W.); 2Joint International Research Laboratory of Agriculture & Agri-Product Safety, Yangzhou University, Yangzhou 225009, China

**Keywords:** pig, PEDV, *AQP3* gene, methylation, Sp1 transcription factor

## Abstract

As an important carrier for intestinal secretion and water absorption, aquaporin 3 (AQP3) is closely related to diarrhea. In this study, we investigated the mechanisms of *AQP3* gene expression regulation in porcine epidemic diarrhea virus (PEDV)-induced diarrhea confirmed by PCR amplification and sequencing. Evaluation of intestinal pathology showed that diarrhea caused by PEDV infection destroyed the intestinal barrier of piglets. qPCR analysis showed that *AQP3* expression in the small intestine of PEDV-infected piglets was extremely significantly decreased. qPCR and Bisulfite sequencing PCR revealed an increase in the methylation levels of both CpG islands in the *AQP3* promoter region in the jejunum of PEDV-infected piglets. The methylation of mC-20 and mC-10 sites within the two CpG islands showed a significant negative correlation with *AQP3* expression. Chromatin Co-Immunoprecipitation (ChIP)-PCR showed that the Sp1 transcription factor was bound to the *AQP3* promoter region containing these two CpG sites. *AQP3* expression was also extremely significantly reduced in Sp1-inhibited IPEC-J2 cells, indicating that abnormal methylation at the mC-20 site of CpG1 and the mC-10 site of CpG2 reduces its expression in PEDV-infected piglet jejunum by inhibiting the binding of Sp1 to the *AQP3* promoter. These findings provide a theoretical basis for further functional studies of porcine *AQP3*.

## 1. Introduction

Porcine epidemic diarrhea virus (PEDV), transmissible gastroenteritis virus (TGEV), and rotavirus (PoRV) are the main causes of porcine viral diarrheal disease through individual or mixed infections [1]. PEDV is a single-stranded positive-strand alpha coronavirus with a length of approximately 28 kb. PEDV infects pig of all ages, causing a series of pathological changes, such as intestinal salt metabolism imbalances, osmotic pressure changes, and inflammation, with symptoms such as acute diarrhea, vomiting, and even death. The mortality rate of PEDV-related diarrhea in piglets less than 2 weeks of age can reach 100%, which is a serious threat to the pig industry worldwide [2,3,4]. PEDV encodes four structural proteins: spike (S), nucleocapsid (N), membrane (M), and envelope (E). Mutations in the S protein are common, forming numerous variant strains of the PEDV virus. These variations cause problems such as enhanced virulence, genetic diversity in different countries, and even between different regions of the same country, which impede the prevention and control of PEDV infection [5]. The M protein is essential for the assembly and budding of the PEDV virus, and it is an important structural protein for immune protection following PEDV infection. Its sequence is highly conserved among different PEDV strains, which is mostly used to identify the infection of PEDV [6]. At present, there are few studies on the resistance function and mechanism of the key genes involved in PEDV infection. Only the porcine *APN* gene has been shown to bind specifically to PEDV, but whether it is the only functional receptor of PEDV infection is still controversial [7,8,9,10]. This seriously limits our understanding of the pathogenic mechanism of PEDV and the acquisition of resistance function genes and their application in pig disease-resistance breeding. Therefore, screening and analyzing the function of key genes related to resistance to PEDV infection will not only expand our understanding of the mechanisms of PEDV infection and regulation of the host immune response but also promote the detection of related molecular targets and the development of genetic resistance strategies for the prevention and control of epidemic diarrheal disease.

DNA methylation is an important epigenetic modification, occurring mainly in CpG-rich island regions. This modification process involves the covalent methylation of 5-cytosine to form 5-methylcytosine and functions to alter the genetic regulatory mechanism without changing the gene sequence [11]. The promoter region is usually rich in CpG sites and susceptible to cytosine methylation. Its spatial structure is changed by methylation, hindering the binding of promoters to transcription factors and functionally inhibiting gene transcription [12]. There is increasing evidence that the methylation of gene promoter regions is involved in the development of Alzheimer’s disease, Parkinson’s disease, and a variety of tumors, playing an important role in disease prediction and treatment [13,14,15,16]. There are also reports of correlations between promoter methylation and gene expression and disease in pigs. Methylation of the porcine *LYN* gene promoter region plays an important role in regulating its transcription [17]. In Sutai *Escherichia coli* F18-sensitive and resistant pigs, *FUT2* methylation changes at the mC-6 and mC-22 sites were significantly negatively correlated with *FUT2* mRNA expression level. Furthermore, methylation at the mC-22 site inhibited the binding of Sp1 to the *FUT2* promoter, thereby reducing *FUT2* expression and enhancing resistance to *E. coli* F18 in weaned piglets [18].

PEDV first invades the intestinal organs of pigs, leading to an imbalance in intestinal water and salt metabolism and causing diarrhea in piglets. Aquaporins (AQPs) are the main proteins responsible for the transcellular transport of water molecules in the intestine. Aquaporin 3, which is a major intrinsic protein (MIP) of the membrane channel body, is a member of the aquaporin family aquaglycerin channel subfamily that is widely present in the digestive and respiratory tracts [19]. AQP3 assembles as a homotetramer on the cell membrane to form funnel-shaped pores, allowing small solutes such as water, glycerol, and urea to rapidly move in and out of the cell across the osmotic gradient [20]. Thus, *AQP3* plays an important role as a carrier in intestinal water absorption and secretion as well as in maintenance of the balance of intestinal water. Abnormal *AQP3* expression causes disturbances in intestinal water metabolism, which in turn lead to the occurrence of diarrheal diseases. The cholera virus has been shown to inhibit *AQP3* expression in the small intestine by causing an increase in cAMP levels, which in turn causes diarrhea [21]. *AQP3* expression in rat villous epithelial cells is decreased during *Escherichia coli*-induced diarrhea [22]. Abnormal *AQP3* expression has also been detected in the colon of rotavirus-infected mice [23]. *AQP3* has also been found to promote cell migration and proliferation as well as regulate the renewal and repair of intestinal epithelial cells [24,25]. *AQP3* also has a very close relationship with the intestinal barrier.

In this study, we investigated the mechanism underlying the regulation of porcine *AQP3* in PEDV-induced diarrhea in piglets from the perspective of methylation, laying a theoretical foundation for subsequent in-depth studies of the function of *AQP3* in PEDV infection.

## 2. Materials and Methods

### 2.1. Experimental Animals and Sample Collection

Ternary crossbred Duroc, Landrace and Large White pigs were selected as the experimental animals. According to the clinicopathologic characteristics, three 8-day-old piglets with diarrhea and three 8-day-old normal pigs under the same feeding conditions were sacrificed humanely by intravenous injection of pentobarbital sodium as necessary to ameliorate suffering. Duodenal, jejunal and ileal tissues, jejunal contents, and fecal samples were collected and stored in liquid nitrogen in situ prior to storage at −70 °C.

This animal study was approved by the Institutional Animal Care and Use Committee (IACUC) of the Yangzhou University Animal Experiments Ethics Committee, China (permit number: SYXK (Su) IACUC 2012-0029; Approval Date: 09-12-2012). All experimental procedures were performed in accordance with the Regulations for the Administration of Affairs Concerning Experimental Animals approved by the State Council of the People’s Republic of China.

### 2.2. Identification of Virus Types in Diarrheal Piglets

Jejunal contents and fecal samples were diluted in 500 μL phosphate-buffered saline (PBS) dilution and then subjected to three freeze–thaw cycles at −20 °C before centrifugation (8000 rpm, 5 min) to obtain the supernatant. Total RNA was extracted from the supernatants using TRIzol reagent (Invitrogen, Carlsbad, CA, USA) according to the manufacturer’s instructions. The nucleic acid concentration was measured with a NanoDrop 1000 (Thermo Fisher Scientific, Waltham, MA, USA) and reverse transcribed using a reverse transcription kit (Vazyme, Nanjing, China) according to the manufacturer’s instructions. Viruses were identified by PCR amplification of 1.0 μL of each of the cDNA samples using primers designed based on the gene sequences for PEDV (AF353511), TGEV (FJ755618), and PoRV (FJ807867) published in the GenBank database. Primer sequences are listed in Appendix A. PCR products were detected by 2.0% agarose gel electrophoresis, and PCR products of the same length as the target fragment were sequenced. Primers were designed using by Primer Premier 5.0 software. The primer synthesis and PCR product sequencing were completed by Sangon Biological Engineering Co., Ltd. (Shanghai, China).

### 2.3. Preparation and Evaluation of Paraffin-Embedded Tissue Sections

Approximately 3 cm of the middle segment of the duodenum, jejunum, and ileum tissues of normal and diarrheic piglets were collected, washed with PBS, and fixed in 4% paraformaldehyde solution for 24 h. Then, the tissues were dehydrated, cleared, and embedded in paraffin. Then, serial sections (thickness approximately 5 mm) were prepared with a microtome and placed in 45 °C water to develop slides, after which the wallet was picked up with a glass slide. The slides were heated at 60 °C for 1 h and then deparaffinized before staining using a hematoxylin–eosin (HE) processing kit (Beyotime Biotechnology, Shanghai, China) according to the manufacturer’s instructions. Paraffin-embedded sections were observed in detail under a light microscope (Olympus, Tokyo, Japan) to compare the changes in the morphological structure of the individual intestinal mucosa.

### 2.4. Real-Time PCR Analysis

Real-time qPCR analysis was conducted on an ABI7500 (Applied Biosystems, Foster City, CA, USA) using the following conditions: 95 °C for 30 s followed by 40 cycles of 95 °C for 5 s and 60 °C for 34 s. The primers for *AQP3* (HQ888860.1) and *Sp1* (XM005652567.3) are shown in Appendix A. *GAPDH* (AF017079.1) and *β-actin* (XM00312428.3) were chosen as housekeeping genes because of their high expression stability in pig tissues [26]. The primers for these two genes are shown in Appendix A.

Three replicates of each sample were included and after the amplification procedure, the lysis curve was analyzed to confirm the specificity of the amplification products.

### 2.5. Methylation Analysis of the Porcine AQP3 Promoter Region

CpG islands in the promoter region 2000 bp upstream of *AQP3* were predicted by using the MethPrimer website, and methylation primers were designed for regions where CpG islands were present. DNA was extracted from jejunal tissue of diarrheic and normal piglets using the TIANamp Genomic DNA kit (Tiangen Biotech, Beijing, China). Extracted DNA (500 ng) was used for bisulfite treatment according to the instructions of the EZ DNA Methylation-GoldTM Kit (Zymo Research, Irvine, CA, USA). The extracted DNA was amplified by PCR using ZymoTaq PreMix (Zymo Research, Irvine, CA, USA) with the two pairs of methylation primers shown in Appendix A.

The 25-μL amplification system contained 3.0 μL transformed DNA, 1 μL for upstream and downstream primers, 12.5 μL Zymo Taq Premix, and RNase-free water. The reaction conditions were as follows: 95 °C for 10 min, followed by 40 cycles of 95 °C for 30 s, 52 °C (CpG2 primer 50 °C) for 30 s, and 72 °C for 35 s, with a final incubation at 72 °C for 10 min. Subsequently, the PCR products were separated by 2% agarose gel electrophoresis to confirm the specificity of the primers, and the target fragment was recovered using a gel recovery kit (Tiangen Biotech, Beijing, China) according to the manufacturer’s instructions. The PCR product was ligated into the pMD19-T vector (TaKaRa, Dalian, China) and transformed into *E. coli* DH-5α (Tiangen Biotech, Beijing, China) before overnight in the presence of ampicillin. Fifteen ampicillin-resistant monoclonal colonies were picked, cultured at 37 °C for 12 h, and sequenced. The upstream sequences of *AQP3* were aligned using QUMA online software to analyze the degree of methylation at each CpG site.

### 2.6. Transcription Factor Binding Domain Prediction and Chromatin Co-Immunoprecipitation (ChiP) Analysis

Alibaba 2 online software was used to predict the transcription factor-binding domain in the -2000 bp promoter region of the porcine *AQP3* gene. Porcine jejunum tissues (80 mg) were cut into pieces (0.5–2 mm^3^), collected in a centrifuge tube containing 1 mL of precooled PBS, mixed with formaldehyde solution, and subjected to chromatin co-immunoprecipitation assay in strict accordance with the instructions for the Pierce Agarose ChIP Kit (Thermo Fisher Scientific, Waltham, MA, USA). Input, immunoprecipitated (IP) and negative control samples were prepared using enzyme-free water, Sp1, and IgG antibodies, respectively. The DNA sequence of the predicted binding region was used as a template to design ChIP-PCR primers and perform PCR amplification; the two pairs of primers are shown in Appendix A.

### 2.7. Sp1 Transcription Factor Interference

The RNAi Designer online website was used to design mRNA for sequence interference targeting the porcine Sp1 transcription factor and negative control shRNA targeting an unknown porcine sequence. The four Sp1 interfering shRNA sequences and negative control shRNA sequence are shown in Appendix A.

The sequences were cloned and assembled into a lentiviral vector expressing GFP by GenePharma Company (Shanghai, China) and co-transfected with packaging plasmid into 293T cells for virus production. IPEC-J2 cells were inoculated into a 12-well plate and cultured in complete DMEM medium containing 10% FBS in a CO_2_ incubator until they reached approximately 80% confluence. Then, 10 L Sp1 interference and 10 L negative control virus solution (1 × 10^8^ TU/mL) were added to the 12-well plate with three replicates in each group; a non-infected group was also set up. After 48 h, the fluorescence of cells in the Sp1 interference and negative control groups were observed. Cells were screened in the presence of 10 mg/mL puromycin. The cell culture medium containing puromycin was changed every 24 h until all normal cells had died. Sp1 interference and negative control cells were collected, and total cellular RNA was extracted for quantitative detection of the relative expression of Sp1 and *AQP3*.

### 2.8. Statistical Analysis

The real-time fluorescence quantification data were analyzed using the 2^-ΔΔCt^ method. Differences in *AQP3* expression levels in the small intestine of diarrheal and normal piglets were analyzed by an independent sample *t*-test based on a normality and homoscedasticity test. Pearson correlation analysis was performed to evaluate the correlation between methylation levels at different CpG sites and gene expression. All statistical analysis was conducted using SPSS 22.0 software. *p* < 0.05 means the difference is significant at the 0.05 level, and *p* < 0.01 means the difference is significant at the 0.01 level.

## 3. Results

### 3.1. Selected Diarrheal Piglets Were Only Infected with PEDV

The PCR products amplified using the PEDV, TGEV, and PoRV primers were 216, 252, and 291 bp in length, respectively. Agarose gel electrophoresis confirmed (Appendix A) the successful amplification of PCR products of the appropriate target sequence length from jejunal tissue, jejunal content, and fecal samples of diarrheal piglets using PEDV-specific primers, whereas no PCR products were obtained using primers specific for the other virus primers. No PCR products were obtained from normal piglet samples. The PEDV-specific PCR products were sequenced, and Blast alignment showed a high degree of similarity with a variety of PEDV-M gene sequences, indicating that diarrheal piglets were infected only with the PEDV virus.

### 3.2. Intestinal HE Staining

Examination of the intestinal mucosa of diarrheic and normal piglets under a light microscope showed that the tissue structure of normal piglets was intact, the villi were arranged neatly, and the outline of the intestinal glands was clear. In contrast, the intestinal villi’s height of diarrheic piglets was significantly reduced, with breakage and partial necrosis and shedding of villous epithelial cells (Figure 1).

### 3.3. Expression of AQP3 in Intestinal Tissues of Normal and Diarrheic Piglets

*AQP3* expression was detected in the small intestinal tissues of diarrheic and normal piglets by qPCR. In normal piglets, *AQP3* expression was lower in the duodenum and higher in the ileum. Compared with normal piglets, *AQP3* expression in the three intestinal segments of diarrheic piglets was significantly downregulated (*p* < 0.01), with the most significant downregulation (approximately 11-fold) in the jejunum (Figure 2).

### 3.4. Prediction of AQP3 CpG Islands and Methylation Analysis

As shown in Figure 3A, MethPrimer prediction revealed that there were two CpG islands, designated CpG1 and CpG2, which were located at −1753 to −1501 bp (length 253 bp) and −56 to −243 bp (length (188 bp), respectively, in the -2000 bp promoter region of the *AQP3* gene (Figure 3A). As shown in Figure 3B and Appendix A, the length of the targets fragment amplified by the two pairs of primers were 300 bp (−1762 to −1463 bp) and 388 bp (−295 to 87 bp), respectively. Methylation analysis revealed that the overall methylation level of the two CpG islands in the *AQP3* promoter region in PEDV-infected piglets was higher than that in normal piglets, although the difference was not statistically significant, with a higher overall methylation level in CpG1 and a very low overall methylation level in CpG2 (Figure 3C). CpG1 had a total of 21 CpG sites, all of which were methylated to varying degrees. CpG2 had a total of 26 CpG sites, although only six were methylated to varying degrees, and the remaining 20 sites were unmethylated (Figure 3D).

### 3.5. Analysis of the Correlation between Methylation Degree and Gene Expression

The correlation between methylation levels and mRNA expression of the CpG sites within the two CpG islands of *AQP3* in normal and diarrheal piglets was analyzed using the Pearson method (Figure 4). There was no significant negative correlation between the change in the global methylation degree of CpG1 and gene expression (*r* = −0.462, *p* = 0.357), of which the change in the degree of methylation of 17 CpG sites was negatively correlated with gene expression, and the mC-20 site showed a highly significant negative correlation with gene expression (*r* = −0.920, *p* = 0.009). There was no significant negative correlation between the change in the global methylation degree of CpG2 and gene expression (*r* = −0.795, *p* = 0.059), of which five sites were negatively correlated with gene expression, one site was positively correlated, and the other sites were unmethylated; furthermore, there was a significant negative correlation between the mC-10 site and gene expression (*r* = −0.869, *p* = 0.025).

### 3.6. Transcription Factor Prediction and Binding Analysis

As shown in Figure 5A, the prediction of transcription factors in the region near the two CpG islands in the promoter region of *AQP3* revealed as many as 14 transcription factors that were bound by the two CpG island sequences, including Adf-1, Sp1, C/EBPalpha, MoyD, and Otc-1. However, both the mC-20 site of CpG1 and the mC-10 site of CpG2, which exhibited a negative correlation with gene expression, were within the Sp1 binding region. The lengths of the two pairs of primer products were 55 bp (-1529 to -1475 bp) and 100 bp (-170 to -71 bp), respectively (Figure 5B). ChIP-PCR analysis showed the two primer sets amplified bands in the Input and IP samples, and the Input band was brighter than the IP band (Figure 5C). There was no band in the negative control IgG sample or the sample containing water as the template, thus excluding the possibility of false positive results. Fluorescent protein expression was detected following the lentiviral infection of IPEC-J2 cells with the four Sp1-interfering shRNAs (Figure 6A). Quantitative detection of the interference efficiency of shRNA in each group revealed that Sp1-521 had the highest interference efficiency at approximately 73% (Figure 6B). As shown in Figure 6B, detection of the relative expression of AQP3 in shRNA interference and negative control IPEC-J2 cells revealed that Sp1 interference significantly reduced *AQP3* expression (*p* < 0.01).

## 4. Discussion

PEDV mainly infects the intestinal epithelial tissues, resulting in intestinal barrier dysfunction in piglets and causing a series of pathological phenomena, such as increased water accumulation in the intestinal lumen, intestinal bleeding, intestinal villous atrophy, cytoplasmic vacuolization, and the shedding of intestinal epithelial cells [27]. In this study, we investigated piglets that were confirmed to be infected with PEDV alone by PCR amplification and sequencing. Evaluation of the intestinal pathology showed disruption of the intestinal villi of piglets infected with PEDV and necrosis and detachment of the intestinal epithelial cells, indicating that the intestinal barrier was damaged. Similar to previous reports, these findings further confirmed that PEDV destroys the intestinal barrier and causes diarrhea in piglets. Subsequent qPCR analysis revealed that *AQP3* expression in the small intestine of PEDV-infected piglets was extremely significantly downregulated compared with normal piglets, with the most significant decrease in the jejunum tissue. Lv et al. reported the same trend of *AQP3* expression in the colon of pigs with diarrhea caused by enterotoxigenic *E. coli* (ETEC) [28]. Studies have shown that *AQP3* expression is competitively regulated by miR-874 and lncRNA H19, which in turn opens the tight junction complex, resulting in decreased occludin and claudin-1 protein expression and increased intestinal permeability [29,30]. In vitro studies of the mucosal barrier of *AQP3* knockout Caco-2 cells compared with normal cells revealed increased *E. coli* susceptibility, decreased transepithelial resistance, and severely impaired cell barrier function [31]. Therefore, it can be speculated that PEDV infection reduces *AQP3* expression and causes damage to the intestinal barrier function in piglets.

To explore the regulatory mechanism of the changes in *AQP3* expression after PEDV infection, the methylation levels of two CpG islands in the *AQP3* promoter region in the jejunum tissues of PEDV-infected and normal piglets were analyzed by methylation sequencing. It was found that CpG1 methylation was very high, and the 21 CpG sites contained were methylated to different extents, while CpG2 methylation was very low, and only six CpG sites contained in it were methylated to a small extent. Studies have shown the existence of CpG sites in the promoter region of each gene, and the methylation at each site will lead to changes in gene expression. It may be that methylation at some sites can silence gene expression, representing a key site for the regulation of gene function, while the methylation of other CpG sites may not affect expression of the corresponding gene [32]. Analysis of the correlation between the methylation level at each CpG site with the mRNA expression level showed that the degree of changes in methylation at the mC-20 site of CpG1 and the mC-10 site of CpG2 was significantly negatively correlated with gene expression. Furthermore, these two sites were predicted to be in the Sp1 transcription factor binding region. Elevated methylation of promoter regions can prevent the binding of positive transcription factors, including Sp1, Myc, and GATA-1, to recognition elements resulting in an inhibition of gene transcription [33,34,35]. There have been many reports that Sp1 transcription factor binding to sequences is regulated by methylation of the gene promoter region [36,37]. In this study, ChIP-PCR analysis showed that the two key CpG sites detected in the promoter region of the *AQP3* gene were located within the Sp1 binding domain. Furthermore, *AQP3* expression was significantly decreased in IPEC-J2 cells with Sp1 knockdown. Thus, we speculated that the increased degree of methylation at mC-20 of CpG1 and mC-10 of CpG2 reduced the binding efficiency of Sp1 to the *AQP3* gene promoter to inhibit *AQP3* expression.

## 5. Conclusions

This study revealed that the downregulation of *AQP3* correlates with PEDV infection in piglets. Two key CpG sites, mC-20 of CpG1 and mC-10 of CpG2, showed a significant negative correlation with gene expression, confirming that Sp1 binds to the region containing these two CpG sites in the *AQP3* gene promoter. We speculate that the increased degree of methylation at mC-20 of CpG1 and mC-10 of CpG2 reduces the efficiency of Sp1 transcription factor binding to the *AQP3* gene promoter and inhibits expression of the *AQP3* gene. Our findings provide insights into the roles of the *AQP3* gene in regulating resistance to PEDV infection and will contribute to its application in genetic breeding strategies against porcine epidemic diarrhea in the future.

## Figures and Tables

**Figure 1 genes-11-01167-f001:**
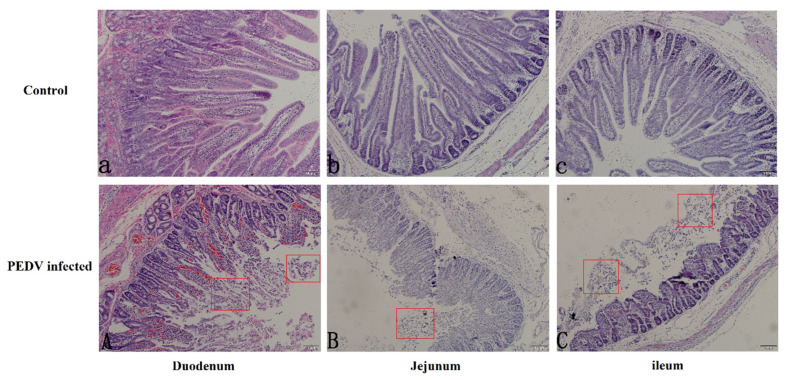
The paraffin sections microscopy of three sections of the bowel in pig intestine (10×). The top group represents the duodenum (**a**), jejunum (**b**), and ileum (**c**) of normal piglets, and the bottom group represents the duodenum (**A**), jejunum (**B**), and ileum (**C**) of diarrheic piglets. The red square marks the area of intestinal villus loss.

**Figure 2 genes-11-01167-f002:**
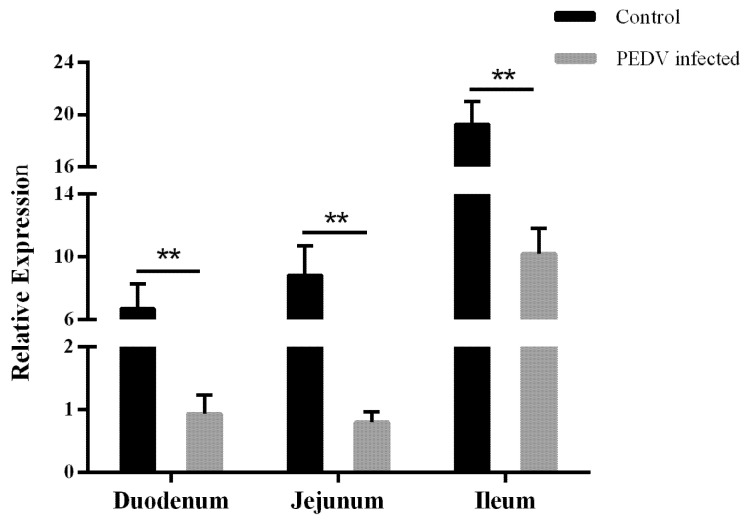
*AQP3* gene expression in gut tissues from porcine epidemic diarrhea virus (PEDV)-infected diarrheic and normal piglets. Bars represent mean±standard deviation (*n* = 3). **, *p* < 0.01.

**Figure 3 genes-11-01167-f003:**
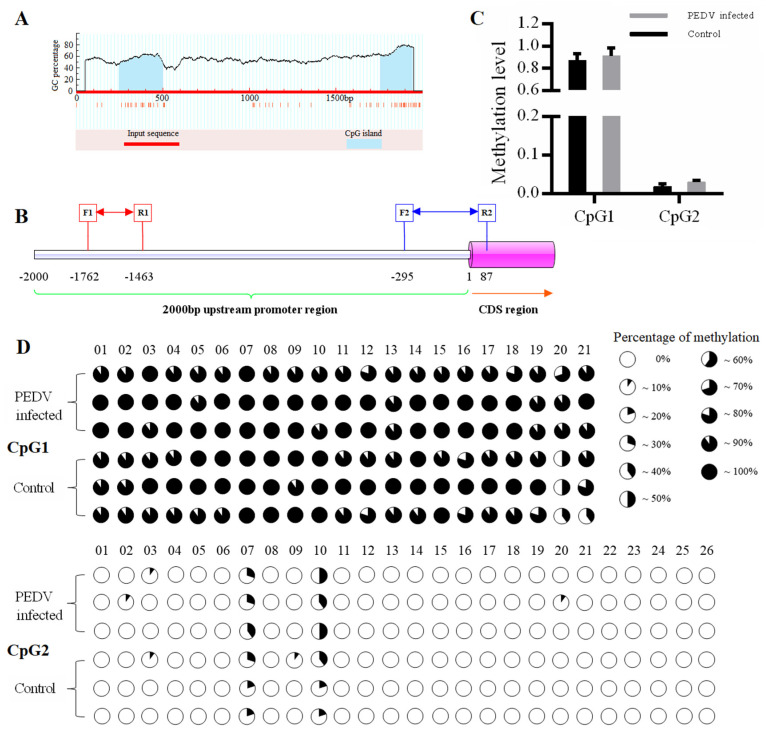
Prediction of CpG islands and methylation status of CpG sites in the promoter region of the porcine *AQP3* gene. (**A**): CpG island prediction in the upstream 2000 bp promoter region of the porcine *AQP3* gene; (**B**): Sequence location of the target fragment of the two pairs of methylation sequence primers; (**C**): Methylation level of CpG islands; CpG1 and CpG2 are the first and second CpG islands, respectively; (**D**): Analysis of the methylation status of CpG sites in two CpG islands in the porcine *AQP3* gene promoter region; the top two groups represent CpG1 and the bottom two groups represent CpG2. CpG sites are marked with pie charts in which the black region represents the methylation level.

**Figure 4 genes-11-01167-f004:**
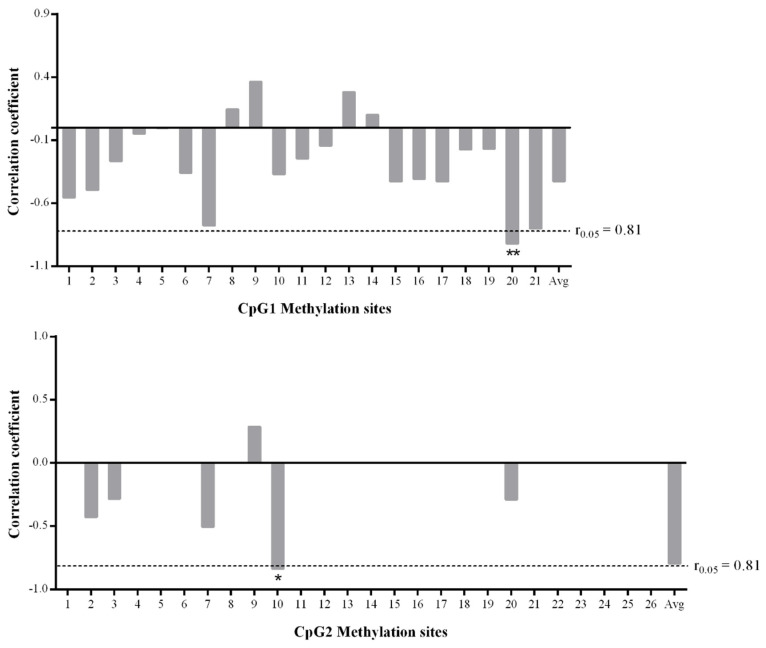
Analysis of correlation between the methylation levels of the CpG sites in the two CpG islands and mRNA expression. 1–21 of the top group mean the sites of CpG1, respectively. 1–26 of the bottom group mean the sites of CpG2, respectively. Avg means the averaged methylation level of all sites in each group (*n* = 3). *, *p* < 0.05; **, *p* < 0.01.

**Figure 5 genes-11-01167-f005:**
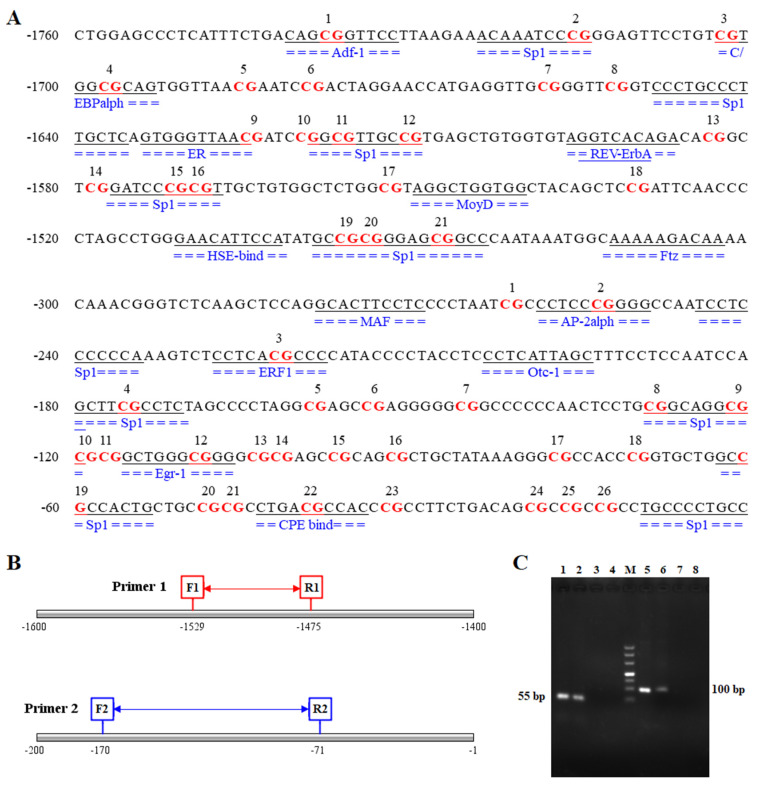
Analysis of the transcription factor binding domain near the CpG island and the result of Chromatin Co-Immunoprecipitation (ChIP)-PCR analysis. (**A**): Sites of transcription factor binding regions near the CpG islands in the porcine *AQP3* gene; (**B**): Sequence position of the fragment targeted by the two pairs of ChIP-PCR primers; (**C**): ChIP-PCR, lanes 1–4 and 5–8 are two different groups representing the PCR products of primer1 and primer2: Input, IP, IgG, and water, M represents the DL500 marker.

**Figure 6 genes-11-01167-f006:**
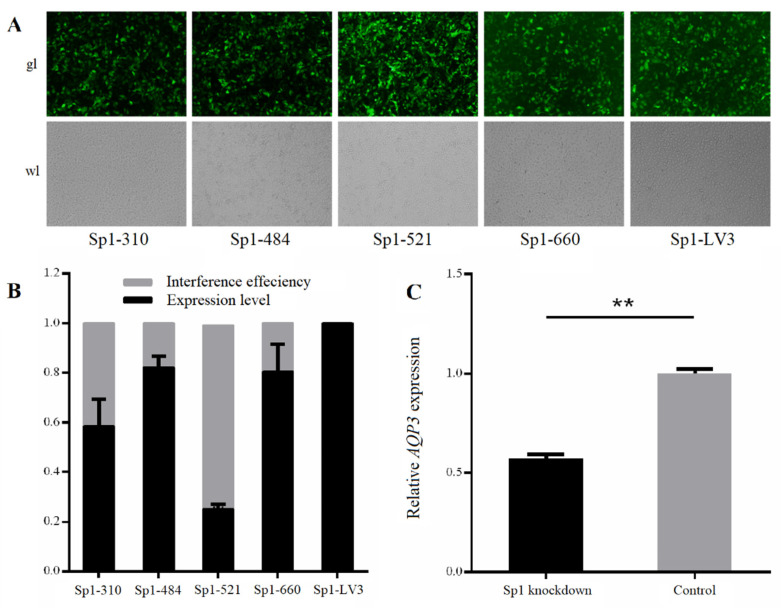
The expression of *Sp1* and *AQP3* genes in IPEC-J2 cells after Sp1 knockdown. (**A**): The white and fluorescence microscope pictures of IPEC-J2 cells transfected respectively with the four Sp1-interfering shRNAs and the control shRNA, represented as Sp1-310, Sp1-484, Sp1-521, Sp1-660, and Sp1-LV3 (200×); (**B**): The interference efficiency of four Sp1-interfering shRNAs; (**C**): The expression of *AQP3* gene in IPEC-J2 cells transfected with Sp1-521 shRNA. **, *p* < 0.01.

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
