# Peer review of "Regulatory Effect of Methylation of the Porcine AQP3 Gene Promoter Region on Its Expression Level and Porcine Epidemic Diarrhea Virus Resistance"

_genes, 2020, doi:10.3390/genes11101167_

Round 1

Reviewer 1 Report

Dear author,

The work studies the mechanisms of AQP3 gene expression regulation in porcine epidemic diarrhea virus, through the analysis of methylation in different CpG island. The paper is novel and interesting from the point of view of response to infection regulation. The manuscript is well structured and gives interesting scientific data about this regulation in pig. However, the authors should implement some minor changes, such as:

Line 91: the authors should include the age of slaughter of the animals.

Line 130: the authors should include a reference indicating the reason for choosing these genes as housekeeping and not others for this species and this type of samples.

Line 179: the authors should include normality and homoscedasticity test performed before analyzing data with t-test.

Line 180: authors indicate that p-values <0.05 have been considered statistically significant. However, p-values <0.01 appear as significant in the figures 2, 4 and 6. Since the p-value is a datum that is defined before performing the statistical analysis, they should unify the information, and consider significant p-values <0.05, as indicated in the methods, or consider significant p-values <0.01, and modify the explanation of how the statistical analysis was carried out.

Line 332: excess tables. The authors could unify the information in the tables S1, S2 and S3.

Reviewer 2 Report

Dear Authors,

I would like to admit, that I read titled ”Regulatory effect of methylation of the porcine AQP3 gene promoter region on its expression level and porcine epidemic diarrhoea virus resistance” with pleasure. Accurate, precise description, broad palette of used molecular methods and essential topic are a vital asset. Viral diseases of farm animals, especially in pigs, highly infectious and with no easy solutions available, can threaten the economy of the agriculture on the national level.

I would like to express my opinion listed for a discussion below:

  • Authors used two groups [three specimens each] for expression analysis and methylation experiments. Is this group large enough to get a picture, possibly the closest to the real situation?
  • Results, paragraph 3.1. There is no approach to determine differences in virus titter/count between three infected specimen. Would it, if discussed, explain in more details observed effects?
  • Results, figure one and others. In my opinion, it would add to clarity if the authors mentioned experimental settings like pooled samples and others, especially for figure 4.
  • Figure 2. Is the different expression of AQP3 in controls (D., J., I.) a biological effect, and if so is there a literature description of this divergence?
  • Figure 3. As far as I understand, there are no differences in methylation pattern between two experimental groups; therefore, its link with EPDV, and especially with EPDV resistance seems elusive. I have an opinion that in proper context, presenting non-significant results is essential and worthy. Therefore, the title of publication might be changed, and authors could pursue a different approach to their otherwise exciting products. 

Kind Regards,

Round 2

Reviewer 2 Report

Dear Authors,

Thank you very much for your reply and explanations. I find your response clear and comprehensive, I agree with your point of view and thank you for the changes made to honour mine.

I have noticed a one last small unclear issue:

In lines 134 – 136 You wrote: „ CpG islands in the promoter region 2,000 bp upstream of AQP3 were predicted by using the MethPrimer website, and methylation primers were designed for regions where CpG islands were present.” Next in line 150 – 151 the text includes the sentence: „The raw AQP3 mRNA sequences were aligned using QUMA online software to analyze the degree of 151 methylation at each CpG site.”

Please clarify which region were subjected to differential methylation study: upstream sequence of AQP3 or mRNA sequence of this gene. Please show the coordinates of investigated CpG region in relation to the gene transcriptional start site.

 After this minor correction I would happily see article published.

Kind regards,